# Bioactive Nutrient Retention during Thermal-Assisted Hydration of Lupins

**DOI:** 10.3390/foods12040709

**Published:** 2023-02-06

**Authors:** Dilini Perera, Gaurav Kumar, Lavaraj Devkota, Sushil Dhital

**Affiliations:** Department of Chemical and Biological Engineering, Clayton Campus, Monash University, Melbourne, VIC 3800, Australia

**Keywords:** hydration kinetics, lupin, mathematical modelling, phytochemical, prebiotic fibre, thermodynamic property

## Abstract

Lupin, an arid pulse, is gaining popularity as a super food due to its superior nutritional properties. However, it has not been considered for large scale thermal processing, e.g., canning. The present work evaluated the best time/temperature combination to hydrate lupins for canning with minimum losses of bioactive nutrients, pre-biotic fibre, and total solids during hydration. The two lupin species showed a sigmoidal hydration behaviour, which was adequately modelled by the Weibull distribution. The effective diffusivity, *D_eff_*, increased from 7.41 × 10^−11^ to 2.08 × 10^−10^ m^2^/s for *L. albus* and 1.75 × 10^−10^ to 1.02 × 10^−9^ m^2^/s for *L. angustifolius* with increasing temperature, namely, from 25 °C to 85 °C. The lag phase decreased from 145 min to 56 min in *L. albus* and 61 min to 28 min in *L. angustifolius*. However, based on the effective hydration rate, reaching the equilibrium moisture, minimum loss of the solids, and prebiotic fibre and phytochemicals, 200 min hydration at 65 °C can be regarded as the optimum temperature of hydration. The findings are thus relevant for designing the hydration protocol to achieve the maximum equilibrium moisture content and yield with the minimum loss of solids (phytochemicals and prebiotic fibres) for *L. albus* and *L. angustifolius*.

## 1. Introduction

Lupin is a member of the Leguminosae family and was domesticated in twentieth century for human consumption and animal feed [1]. *Lupinus angustifolius* (sweet lupin) is a widely cultivated lupin species, followed by *Lupinus albus* (white lupin) and *Lupinus luteus* (yellow lupin) [2,3,4]. Lupins contain high levels of protein (30–44%) and non-starch polysaccharides (40%) and low levels of fat (6–8%) [5,6,7]. The major phenolic chemicals found in lupin are classified as flavones, phenolic acids, and isoflavones. For example, *L. angustifolius* contains 76%, 19%, and 4% flavones, phenolic acids, and isoflavones, respectively [2,8,9]. Variations in phenol content can exist for different cultivar and growth conditions. Lupin is becoming more popular due its nutritional profile and has recently attracted much attention towards the development of lupin-based products. The presence of alkaloids, most notably quinolizidine (in bitter lupins, 1.0–4.5 g per 100 g), limits the consumption of lupins despite their nutritional benefits [6,10]. Health authorities in the UK, Australia, New Zealand, and France have set 200 mg/kg as the amount maximum of quinolizidine that can be found in lupin flour and feed [11,12]. Commercially available cultivars with a low alkaloid content (*L. angustifolius*, *L. albus*, and *L. luteus*) are suitable for use in food and animal feed [13,14].

Numerous medical and animal studies have proven the nutraceutical effect of lupins in reducing obesity, diabetes, and cardiovascular disease [5,15,16]. Lupin has been used to make a variety of food products, including muffins [17], bread [14], noodles [18,19], and pasta [20,21] because of its nutritional and functional value. However, it has not yet been considered for larger scale thermal processing such as canning.

Hydration is a critical and primary unit operation in legume processing. The main purpose of soaking dry legumes before food preparation is to facilitate and improve heat and mass transfer during cooking [22,23]. In addition, it helps to maintain uniform texture throughout the products and removes certain anti-nutritional factors [24]. A previous investigation found that the tropical legume Mucana, which is native to Africa, reduced phytic acid by 27.9% and 36.0% after 6 and 24 h of soaking, respectively, at ambient temperature [25]. Phytase activity increased as a result of the soaking, which decreased the amount of phytate that was present in the grains.

However, hydration is not a simple process; it involves multiple steps such as water absorption, capillary flow, diffusion, and solid matrix relaxation. Similarly, several intrinsic factors such as seed size, seed coat thickness, cotyledon chemical composition, and the size of the micropyle and hilum and extrinsic factors such as the temperature of the soaking medium, pH, solids in media, etc., affect the hydration rate [26,27,28]. Generally, the seed coat of Fabaceae family grains consists of a more complex structure compared to the Poaceae family, such as wheat, barley, corn, and rice [29,30]. The pericarp in Poaceae family grains are very permeable to water, whereas the seed coat of grains from the Fabaceae family can be partially or entirely impervious depending on composition, variety, and moisture [31].

Typically, legume hydration can be conducted for 20–40 min at high temperatures (82–100 °C) or 8–16 h at room temperature [22,32]. Even though hydration at ambient temperature (22 °C) is time-consuming, it is preferred in terms of the nutritional perspective [22,33,34,35]. Elevated temperature is preferred in industrial processing since it increases the water uptake rate and minimises the processing time, allocation of space, tank capacity, and risk of microbial growth. However, elevated temperatures result in low equilibrium moisture content (EMC) and adversely affect thermolabile phenols, anthocyanins, and higher leaching of total solids and prebiotic fibre [33,36]. For example, polyphenols are sensitive to high temperatures, which significantly affect the phenolic content of lupins during hydration. Therefore, optimisation of hydration time/temperature on lupin can provide valuable insights while designing hydration protocols, which maximise the moisture content and reduce the loss of bioactive nutrients, total solids, and prebiotic fibre.

Though legume hydration has been widely investigated, there is still a lack of studies on total solid and phytochemical loss during hydration. Several kinetic models have been developed to estimate the hydration level under various conditions [37]. Among them, the Peleg model, which explains the downward concave behaviour of hydration, is an imperial model to estimate the moisture content at different time intervals of hydration [31]. The Kaptso and Weibull distribution models explains the sigmoidal behaviour of hydration and can predict moisture and initial lag phase time [31]. Most of the previous studies on lupin hydration focused on understanding the hydration behaviour instead of investigating the thermodynamics and the loss of valuable phytonutrients, oligosaccharides, and soluble solids and hardness changes during hydration. In the present work, we analysed the hydration and thermodynamic behaviour of two lupin species: *L. albus* and *L. angustifolius*. Moreover, we elucidated the effects of the hydration time–temperature combination on hardness, total phenols, anthocyanins, probiotic fibre loss, and total solid loss at different time points during hydration at 25 °C, 45 °C, 65 °C, and 85 °C. Therefore, our findings are novel in this regard and have scientific as well as industrial relevance.

## 2. Materials and Methods

### 2.1. Materials and Chemicals

*L. albus* and *L. angustifolius* were supplied by Seednet, Horsham VIC 3400, Australia. *L. albus* and *L. angustifolius* had initial moisture contents of 9.188 ± 0.57 and 10.128 ± 0.3 (g/100 g db.), respectively. Before the experiment, samples were kept at room temperature in an airtight sealed container. Potassium chloride, gallic acid, Folin reagent, sodium acetate, sodium carbonate, oligosaccharide standards, D-stachyose hydrate, raffinose pentahydrate, verbascose, and HPLC grade methanol were purchased from Sigma Aldrich, Melbourne, VIC, Australia.

### 2.2. Physical Properties

Physical properties such as mean diameter (De), volume (*V*), and surface area (*S*) of seeds were measured using the equations reported by Mohsenin [38]. On hundred randomly selected seeds were weighed in triplicate, and the total weight was multiplied by ten to measure the thousand grain weight. A digital calliper was used to measure the width (*W*), length (*L*), and thickness (*T*) of 50 seeds; both types of lupins were considered trapezoid (flat, rectangular, or square in shape, with rounded corners). Then the surface area was calculated using Knud Thomsen approximation combined with Equation (3) [39,40].
(1)$$e={\left(L\times W\times T\right)}^{1/3}$$
(2)$$V=\frac{1}{6}\pi \times L\times W\times T$$
(3)$$S\approx 4\pi \times \sqrt[{1.6075}]{\frac{{\left(L\times \frac{W}{4}\right)}^{1.6075\;}+\;\;{\left(L\times \frac{T}{4}\right)}^{1.6075\;}+\;\;{\left(T\times \frac{W}{4}\right)}^{1.6075\;\;\;\;}}{3}}$$
(4)$$\mathrm{Specific}\;\mathrm{surface}\;\left(\;S_{sp}\right)=\frac{S}{V}$$

### 2.3. Hydration Kinetics and Modelling

The hydration behaviour of lupins was examined using 80 g of samples kept in a water bath (Thermoline TWB-22T) at the different temperatures of 25 °C, 45 °C, 65 °C, and 85 °C (±2). The weight of samples was measured at different time intervals (15 min) by taking out lupins from the mesh bag and blot-drying them using paper towels. Then dry basis moisture content and moisture ratios were plotted against hydration time.

Mathematical modelling was carried out to uncover the hydration behaviour and underlying factors affecting the mass transfer mechanisms. Experimental results were fitted to a mechanistic model and three empirical models. Mechanistic model, based on Fick’s law of diffusion with a lumped parameter, *viz*, effective diffusivity, *D_eff_*, was used. Three widely used empirical model, i.e., Peleg’s model, the Kaptso model, and the Weibull distribution model, were checked for adequacy. Non-linear regression analysis was carried out for fitting the experimental values to model values with the Levenberg–Marquardt iteration algorithm in Origin 18 software package.

#### 2.3.1. Fickian Diffusion Model

Fick’s law of diffusion is the most commonly applied mass transport model in food processing. The mass transfer rate is directly proportional to the curvature of the concentration gradient, according to Fick’s equation of diffusion, and may be expressed mathematically as
(5)$$\frac{\partial c}{\partial t}\propto \frac{\delta^{2}c}{\partial x^{2}}$$
where the term right hand term indicates the rate of concentration change, while the left-hand term is the second partial derivative of concentration with respect to position in the *X* coordinate.
(6)$$\frac{\partial C}{\partial t}=D_{eff}\left(\frac{\partial^{2}C}{\partial x^{2}}\right)$$

Proportionality constant, *D_eff_* (m^2^/s), i.e., effective diffusivity, is the measure of mass transfer in the system. For spherical objects, Equation (6) can be reduced to [41]
(7)$$MR=\frac{M_{e}-M_{t}}{M_{e}-M_{i}}=\frac{6}{\pi^{2}}\overset{\infty }{\underset{i=1}{\sum }}\frac{1}{k^{2}}\;exp\left(\frac{-k^{2}\pi^{2}D_{eff}t}{r^{2}}\right)$$
where *M_e_*, *M_i_*, and *M_t_* are equilibrium moisture content, initial moisture content, and moisture at any time *t*, and *r* is the equivalent radius of the lupin grains. Many researchers have observed that as processing time increases, the infinite series converges rapidly to the first term of the equation, thus taking the first term into consideration and omitting the others [41,42].
(8)$$MR=\frac{6}{\pi^{2}}\;exp\left(\frac{-\pi^{2}D_{eff}t}{r^{2}}\right)$$

Equation (8) was used in the present study to model the hydration behaviour of lupin seeds. A plot between Ln (*MR*) and hydration time was conducted, and the *D_eff_* was determined from the slope of the graph employing the following formula:(9)$$D_{eff}=\;\frac{Slope\;*\;r^{2}}{\pi^{2}}$$

To determine the relationship between *D_eff_* and temperature, an Arrhenius type equation was used. A plot between Ln (*D_eff_*) and 1/*T* was used to determine the activation energy and pre-exponential factor.
(10)$$D_{eff}=D_{0}exp\left(-\frac{E_{a}R}{T}\right)$$

#### 2.3.2. Peleg’s Model

Peleg’s model is a two factor non-exponential model commonly used to demonstrate the downward concave shape hydration behaviour [42].
(11)$$M_{t}=M_{i}+\frac{t}{K_{1\;\;\;}+K_{2}t}$$
where the parameter *K*_1_ is indicative of the water absorption rate, and *K*_2_ is related to the achievable equilibrium moisture content.

#### 2.3.3. Sigmoidal Model

Typically, this model simulates the sigmoidal hydration behaviour and accounts for the lag in water uptake due to the outer seed coat of pulses. The modelling equation is as follows:(12)$$M_{t}=\frac{M_{e}}{1+exp\left[-k.\left(t-\tau \right)\right]}$$
where parameter *k* is related to the water uptake rate, and term *τ* describes the lag phase.

#### 2.3.4. Weibull Distribution Model

The Weibull distribution model, a widely used probability distribution function, has been employed by several scientists to model the hydration and dehydration process with significantly high accuracy. For hydration, the model defined over two-factors, namely, scale parameter α and shape parameter *β*, can be used (Equation (13)) [42].
(13)$$\frac{M_{i}-M_{t}}{M_{i}-M_{e}}=1-exp^{\left(-{\left(\frac{t}{\alpha }\right)}^{\beta }\right)}$$
where all the symbols have the usual meaning as specified above unless stated otherwise.

#### 2.3.5. Determination of Model Suitability

Finally, the model fitting quality was assessed by calculating the coefficient of determination (*R*^2^) and the reduced chi square and root-mean-square error values (*RMSE*).
(14)$$R=\frac{\sum_{i=1}^{N}{\left(Y_{ei}-Y_{p}\right)}^{2}-\sum_{i=1}^{N}{\left(Y_{ei}-Y_{pi}\right)}^{2}}{\sum_{i=1}^{N}{\left(Y_{ei}-Y_{p}\right)}^{2}}$$
(15)$$\mathrm{Reduced}\varkappa^{2}=\frac{1}{N-n}\sum_{i=1}^{N}\frac{{\left(Y_{ei}-Y_{pi}\right)}^{2}}{\sigma_{1}^{2}}\;$$
(16)$$\mathrm{RMSE}=\sqrt{\frac{\sum_{i=1}^{N}{\left(Y_{ei}-Y_{pi}\right)}^{2}}{N}}\;$$
where *Y_ei_* is the experimental value for the *i*th term, *Y_pi_* is the predicted value obtained from the model, *Y_p_* is the mean of experimental values, *N* is the total observations, and *n* is the number of terms in the predicting model.

### 2.4. Thermodynamic Characterisation

The Fickian diffusion model was used to estimate the thermodynamic properties, viz, enthalpy of activation (ΔH), entropy of activation $\left(\Delta S\right)$, and Gibb’s free energy$\;\left(\Delta G\right),$ for the effective diffusivity. These thermodynamic properties were estimated (Equations (17)–(19)) as a function of hydration temperature [39,43,44].
(17)$$\Delta H=E_{a}-RT$$
(18)$$\Delta S=R\left[lnD_{0}-ln\left(\frac{k_{b}}{h_{p}}\right)-lnT\right]$$
(19)ΔG=ΔH−TΔS
where *R* is the universal gas constant (8.314 J/mol.K), *T* is the absolute temperature (K), *k_b_* is Boltzmann’s constant (1.38 × 10^−23^ J/K), and h_p_ is Planck’s constant (6.626 × 10^−34^ J·s).

### 2.5. Morphological Characteristics of Lupins

A scanning electron microscope (SEM) (Phenom XL Phenom World, The Netherlands, Eindhoven) was used to characterise the surfaces of the seed coat, cotyledon, and hilum at 10 kV. The lupin outer surface, seed coat, cotyledon, and hilum with micropyle were cut into thin slices using a scalpel blade and coated with 5 nm platinum before observing under SEM.

### 2.6. Hardness Measurement

To measure the hardness, 80 g of each lupin sample was kept in a mesh strainer bag and maintained under the same hydration conditions. At each sampling point, 20 seeds were taken out, and hardness was measured as a penetration test using a TA.XtplusC texture analyser with a load cell of 5 kg. A needle (P/2) was used to perforate the grain at 0.2 mms^−1^ on the abaxial side (*y*-axis) until a distance of 3 mm was reached.

### 2.7. Total Solid Loss

The solid content of hydration media was measured at several time points during hydration to determine solid loss during hydration. Then the collected samples were freeze-dried, and solid loss was calculated.

### 2.8. Total Phenolic and Total Anthocyanin Loss

Total phenolic content was quantified using the Folin–Ciocalteau method adapted to the micro assay following the method of Devkota et al. [45]. Initially, the sample and Folin reagent (50 μL of each) were added. Then, 100 μL of 0.7 mol dm^−3^ Na_2_CO_3_ was added and incubated for 1 h. Gallic acid was used to create a calibration curve. Using a Tecan Infinite 200PRO microplate reader, the absorbance values of the samples were measured at 765 nm.

The total anthocyanin loss of hydration medium was determined at each time point using the method described by Devkota et al. [45]. KCl buffer (pH 1.0) or sodium acetate buffer (pH 4.5) were used to dilute the samples. Once the dilution factor was determined, samples were mixed with KCl buffer, and absorbance was measured at 520 nm and 700 nm using a Tecan Infinite 200PRO microplate reader (Mannedorf Switzerland).
(20)$$Anthocyanin\;content=\left(C3GE\;\frac{mg}{L}\right)=\frac{A\times MW\times DF\times 10^{3}}{\in \times l}$$

*A* is absorbance difference ((A_520_–A_700_) _pH 1.0_ − (A_520_–A_700_) _pH 4.5_), *l* is the path length in cm, *DF* is the dilution factor, *ε* is the molar coefficient (26,900 in l/mol cm), and *MW* is the molecular weight (449.2 g/mol) of C3G.

### 2.9. Oligosaccharide and Soluble Fibre Loss

The oligosaccharide loss during hydration was measured by HPLC-RI, as described by Devkota et al. [46]. Samples were first filtered using a solid-phase adsorption cartridge and then collected in glass vials after filtering through a nylon filter (0.2 μm). An Agilent HPLC-RI system (Agilent infinity 1260) equipped with an Agilent Hi-Plex Na oligosaccharide column (300 × 7.7 mm) and a Hi-Plex Na guard column (50 × 7.7 mm) was used at a flow rate of 0.2 mL/min. Columns were maintained at 80 °C, and the RI detector was maintained at 45 °C to detect oligosaccharides. A calibration curve was created by using raffinose, stachyose, and verbascose standards.

### 2.10. Data Analysis and Statistics

All tests were conducted in triplicate, and the results were reported as an average. SPSS statistics was used for statistical analysis (IBM, Version 27, 2021). Data were analysed using one-way ANOVA, and mean differences were evaluated using Tukey’s multiple comparison tests at a *p* < 0.05 significance level. The non-linear regression analysis was carried out in Origin 2018 software.

## 3. Results and Discussion

### 3.1. Physical Properties

The physical property attributes of legumes had pronounced effects on their hydration kinetics, chemical, and mechanical behaviour. The seed size, shape, and thickness and the morphology of the seed coat played a vital role in legume hydration since all these factors contributed to the mass and heat transfer through the seed coat [42,47,48]. The physical property attributes for *L. albus* and *L. angustifolius* seeds are shown in Table 1.

Traditionally, *L. albus* are larger in size, volume, and weight than the *L. angustifolius* species, as can be seen in Figure 1 and Table 1. The variation in seed weight can be due to differences in gene pools, storage conditions, and other climatic and agronomical factors [49]. The seed coat thickness of both species was recorded as 0.21 mm ± 0.01. Differences in the porosity of the seed coat and thickness often have significant impacts on legume water uptake [48]. Faster hydration rates positively correlate with thinner seed coats [49]. However, no significant difference was observed between the seed coat thickness of *L. albus* and *L. angustifolius* (*p* > 0.05).

### 3.2. Microstructural Characterisation of Lupin Seeds

Seed microstructures, especially the seed coat, hilum, and micropyle play vital roles in the overall hydration process. According to Garnczarska et al. [50], the hilum and micropyle are the principal openings for water uptake in lupin seeds. Microscopic images show that either species of lupins have a hair rim surrounding the hilum (Figure 2a,f). As expected, the hilum for *L. albus* was greater in size compared to that of *L. angustifolius*. Figure 2b,g present interesting pictures of the porous micropyle recorded at high magnification (2300×). Several tube shaped microchannels can be seen extending from outside of the seed all the way to the inside. There is a high likelihood that these structures are the ones responsible for the capillary transfer of water inside the seed during hydration. The outer surfaces of the seed coats of both species are shown in Figure 2c,h. Although the seed coat might appear smooth to the naked eye, a closer look employing SEM images revealed different ornamental characteristics for each species. For *L. albus* the structure looked like a series of various ridges arranged haphazardly, closely resembling satellite images of a mountainous terrain, while, *L. angustifolius* had plateau-like shapes with flat tops.

Imaging the seed coat cross-section revealed that both species have similar tissue sequences. As can be seen in Figure 2d,i, there are three main cell layers in the lupin seed coat, just like other legumes. The external layer is palisade tissue made up of dried cells and having different hydrophobic substances such as lignin polysaccharides, pectin, suberin, cutin, calose, phenols, and quinones [27,51]. The surface of this layer has an abundance of cuticle compounds of wax that restrict moisture permeability to the grains. It could be the principal cause of the initial resistance to water intake. The second layer is composed of bone-shaped osteosclerosis cells with large intercellular spaces. The third layer is parenchyma, which is made up of several flat layers and is significantly more vulnerable to moisture absorption. The water that flows in through the hilum and micropyle first comes into contact with the parenchyma layer at the cotyledon–hull interface [41,51]. Similar structural arrangements were reported for carioca beans [52].

Both lupin species had similar cotyledon structures, as shown in Figure 2e,j. The cotyledons were highly porous with no starch granular structures, possibly due to the very low amount of starch present in lupin seeds. The cotyledon structures of lupin were very different from those reported for chickpea, faba bean, field peas, and lentils by Jeganathan et al. [53]. In these images, large starch granules imbedded in the protein matrix and jutting out of cell block lets were observed. Morphological analysis conducted on pea, common beans, lentils, and chickpea flour [54] also produced similar results. Lupin cotyledon structures are unique in this regard.

### 3.3. Hydration Kinetics and Modelling

Determination of hydration behaviour is an important consideration for legume processing. The two most common seed grain hydration behaviours are the downward curve shape (DCS) and sigmoidal [28,40]. The former is commonly observed for cereal grains and the latter for pulses. Both lupin species exhibited sigmoidal hydration behaviour, as exhibited in Figure 3. Similar results were reported for common beans [39], navy beans [42], and mung beans [27]. The outer hull layer restricts the rapid moisture migration into the seeds, resulting in a lag phase, and leads to sigmoidal hydration behaviour. Miano and co-workers suggested that during the lag phase, mass transfer takes place through the hilum and micropyle. Once sufficient water is absorbed by the grain, the outer hull layer gradually transitions from the glassy to the rubbery state and allows for rapid moisture uptake [27,39].

The equilibrium moisture content (EMC) is the measure of maximum moisture a seed can hold upon reaching equilibrium with the surroundings [52]. Temperature greatly affected EMC and the hydration time. For *L. albus*, an initial increase in temperature resulted in higher EMC; however, any further increase caused a reduction in EMC. Likewise, for *L. angustifolius*, an increase in temperature caused a decrease in EMC. The initial increase in EMC for *L. albus* may be due to the expansion of intercellular spaces and pores at medium temperatures, but at elevated temperatures high soluble solid loss may have led to a reduction in EMC [55]. The reduction in EMC may also attributed to the denaturation of the lupin protein matrix, which can decrease the numbers of water bonding site, effectively reducing the equilibrium moisture content.

#### 3.3.1. Fickian Diffusion Model

Table 2 shows the parameter estimates for Fick’s model. It was observed that at high temperatures, the model showed good fit; however, at lower and moderate temperatures, the fit was not satisfactory. Fick’s law of diffusion makes various assumptions that do not exist in physical reality and may be the possible cause for such observations [41,42]. For instance, according to Fick’s law of diffusion, the only mode of mass transfer is diffusion, while in practice, capillary mass transfer, vapor pressure difference, and several other factors are involved in mass transfer. Similarly, uniform heat and mass transfer and isotropy are unattainable constraints. Despite these limitations, Fick’s law provides good estimates of mass transfer rates and is widely used across the food industry.

*D_eff_* indicates the combined effect of all modes of mass transfer but does not provide much mechanistic insights into the phenomena. *D_eff_* values for *L. albus* and *L. angustifolius* were in the range 7.41 × 10^−11^ to 2.08 × 10^−10^ m^2^/s and 1.75 × 10^−9^ to 1.02 × 10^−9^ m^2^/s, respectively. The rate of moisture uptake was higher for *L. angustifolius*. Higher mass transfer rates in *L. angustifolius* may be due to the smaller size and higher effective surface area. The high fat content in *L. albus* may also inhibit faster moisture uptake, hence lowering the *D_eff_* value [52]. Moreover, Figure 2g,h highlight the structural difference in the seed coat composition of both lupin species, which may lead to the difference in the *D_eff_*. Wang and co-workers reported on similar lines for four different soybean varieties, where the compositional differences in the seed coat caused differences in hydration rates [56]. The hydration temperature had a direct effect on the *D_eff_* of both lupin species, and an increase in temperature increased the *D_eff_*. High temperatures impart higher kinetic energy to the water molecules, which can increase intermolecular collision and help in faster water absorption [57]. Moreover, high hydration temperature can trigger cell wall deterioration due to the depolymerisation of cellulose and pectin. This can cause development of cracks on the seed coat and can facilitate faster mass transfer [58].

The activation energy (E_a_) values of *L. angustifolius* and *L. albus* were 25.52 kJ/mol and 16.00 kJ/mol, respectively (Table 2). The values were estimated using an Arrhenius type equation with a significantly high fit between the experimental and estimated data, i.e., (>0.90). A high E_a_ for *L. angustifolius* indicated that the energy requirement to initiate the hydration process for this was higher compared to the *L. albus*. A high E_a_ also signified that the *D_eff_* for *L. angustifolius* had high thermal sensitivity, and that a slight change in temperature can cause large changes in the *D_eff_* values [59]. A pre-exponential factor signifies the collision rates between molecules. With an increase in temperature, an increase in the pre-exponential factor of both lupin species was observed, indicating that temperature increases accelerated the molecular collision rates and facilitated faster formation of the activated complexes [59].

#### 3.3.2. Peleg’s Model

Table 3 shows the estimated values of parameters obtained after non-linear regression analysis. The Peleg model exhibited the least fit with R^2^ values of 0.94–0.98 and 0.94–0.99, reduced ϰ^2^ values of 11.33–145.34 and 9.84–32.31, and *RMSE* values of 9.66–15.87 and 5.47–11.54 for *L. albus* and *L. angustifolius*, respectively. High reduced *ϰ*^2^ and *RMSE* values signified high residuals and a lack of fit, which may be due to the simplistic nature of the equation and non-consideration of hydration lag phenomena, which was evident for lupin hydration. The parameter *k*_1_ indicates the rate of hydration; a high *k*_1_ value indicates lower mass transfer and vice versa. The *k*_1_ value was found to be higher for *L. albus* compared to *L. angustifolius*, implying a higher rate of mass transfer in the latter variety. In addition, as the hydration temperature increased, the *k*_1_ values decreased. The parameter *k*_2_ indicates the EMC characteristics, and an increase in its value signifies a lowering of the EMC and vice versa. For *L. albus*, at medium temperatures (up to 65 °C), the *k*_2_ value was lower compared to values at 25 °C; however, at high temperatures (85 °C), the *k*_2_ value was higher. For *L. angustifolius*, the values started increasing with temperature. These conflicting results outline the complex nature of the hydration process, which depends on various intrinsic and extrinsic factors.

#### 3.3.3. Sigmoidal Model

The sigmoidal model performed fairly well in terms of predicting the lupin hydration behaviour model with *R*^2^ > 0.97, reduced *ϰ*^2^ < 71.63, and *RMSE* < 7.78. The value of *k* and *τ* decreased with an increase in hydration temperature. Elevated temperatures caused a many-fold reduction in the *τ* value, meaning a decrease in the lag phase time. As outlined in previous sections, this may be due to pectin and cellulosic depolymerisation, leading to the development of fractures in the seed coat [58]. Moreover, higher temperatures accelerate the moisture uptake by the seeds and decrease the resistance to moisture influx, which may lead to reductions in the lag phase [26,60]. Similar observations were reported for mung bean [27], adzuki bean [26], and cowpea [60].

#### 3.3.4. Weibull Distribution Model

The Weibull distribution model predicted the hydration characteristics of both lupin species with distinctively highest accuracy, i.e., *R*^2^ > 0.99, reduced *ϰ*^2^~0, and *RMSE* < 0.03 for all the observations across the temperature range. The α value is inversely proportional to the rate of mass transfer. As shown in Table 3, *α* and *β* values for both lupin species decreased at higher temperatures, implying higher mass transfer rates at elevated temperatures.

### 3.4. Thermodynamic Properties

Table 2 displays the thermodynamic properties of both *L. albus* and *L. angustifolius* hydration processes. The lupin hydration process was analysed employing the transition state theory. Thermodynamic properties of the hydration process were estimated to develop insights into the nature of reactions taking place during the hydration process. Effective diffusivity was considered as the rate parameter for all the calculations. The enthalpy of activation (Δ*H*) for both lupin species was positive, implying that the reaction was endothermic during the formation of the activated complex. *L. albus* had lower values of activation enthalpy, indicating that the energy requirement for excitation of *L. albus* from the ground to the transition state was lower than that of *L. angustifolius*. It is noteworthy that even though the rate of hydration was lower for *L. albus*, thermodynamic analysis revealed that it is easier to hydrate with lower thermal energy consumption. Likewise, a higher temperature caused a reduction in the Δ*H* values, indicating that at elevated temperatures, activated complexes can be formed with lower energy requirements. This is natural, as at high temperatures, the kinetic energy in the system is higher, so molecules have high mobility and frequent collisions [59,61].

Entropy of activation (Δ*S*) values for both lupin varieties had negative signs, signifying an organised molecular structure with limited degrees of freedom [59,61]. As two reactant species, i.e., lupin seed and water in our case, collide with each other to form an activated complex, they undergo orientational distortion, resulting in a change in molecular bond angles and intermolecular distances. Finally, the two species aggregate to form one complex, which naturally restricts its rotation and translation [39,43]. Similar observations were reported for the hydration of soybean [62], common bean [39], and faba beans [63]. As can be seen in Table 2, *L. angustifolius* had lower Δ*S* values, signifying higher molecular ordering.

Gibbs free energy (Δ*G*) values were observed to be positive for both lupin species and indicated that the process is non-spontaneous. Similar to our observation, Borges et al. [62] reported non-spontaneity for soybean hydration and suggested that Δ*G* can also be a measure of the driving force dictating the hydration process. Miano et al. [39] also observed similar trends for the hydration of common beans. The thermodynamic properties outline that the hydration temperature plays a minimal role in activated complex formation and molecular structuring during the hydration process. Factors such as composition and morphology of a seed have more pronounced effects on their thermodynamic properties.

### 3.5. Changes in Hardness during Hydration

Grain hardness is an important factor that influences the processability and palatability of grains. Figure 4a,b demonstrate hardness for both species, i.e., *L. albus* and *L. angustifolius*, as a function of time and its temperature dependence. We observed that *L. albus* had higher initial hardness (55.54 N) compared to *L. angustifolius* (34.94 N), which can be due to the difference in gene pools, sizes of the seeds, storage conditions, and agronomic factors [64]. As expected, a significant reduction in lupin hardness was seen with an increase in the hydration temperature. For instance, the hardness of EMC for *L. albus* hydrated at 25 °C was 2.15 N, which decreased to 0.59 N as the hydration temperature was elevated to 85 °C. Similarly, for *L. angustifolius*, the hardness decreased from 2.34 N to 1.87 N upon increasing the temperature from 25 to 85 °C. This can be further confirmed by considering the diffusivity values. At a high temperature (85 °C) D_eff_ was significantly higher than at a low temperature (25 °C) in both lupin species, which explains the higher moisture transfer during hydration that caused seed softening. Moreover, structural differences in the seed coat of both lupin species may lead to differences in the hardness. Wang et al. [56] reported similar results for four different varieties of soybeans. Likewise, other leguminous seeds such as adzuki beans [65] and cowpeas [55] exhibited softening upon hydration and with an increase in temperature. Possibly, these changes are due to pectin degradation, which in turn dilutes the intercellular layer and cell wall polysaccharides and results in weaker structural integrity [66].

It is interesting to note that the change in grain hardness did not follow a linear trend, as can be clearly seen Figure 4a,b. Corelating the hardness kinetics with the hydration kinetics graph (Figure 2), it was revealed that during the lag phase of hydration, the harness plateaued and did not change much. After the lag phase, once the mass transfer rate increased, the hardness values dropped exponentially. This trend was the same for both the species and the temperature conditions. High hydration rates are indictive of high-water uptake in the lupin grains, which can render them soft and a cause of such observations. As seen in Figure 4a,b, hardness values were close at different hydration values at 85 °C and 65 °C. Therefore, it is plausible to conclude that hydration at 65 °C for less than 200 min is best for obtaining the optimum desired texture of lupin seeds.

### 3.6. Total Solid Loss during Hydration

Hydration at elevated temperatures destroys most of the thermolabile phytonutrients and accelerates leaching of phytochemicals, soluble solids, and prebiotic fibres into the hydration media [46,58]. Therefore, it is critical to understand the leaching behaviour of these components at different time points. Figure 4c,d illustrate the solid loss during hydration at different temperatures for both lupin species. The results show that *L. angustifolius* has significantly higher solid leaching than *L. albus* at all temperatures. This may be due to the small seed size, high surface area, seed coat surface, and internal structure of *L. angustifolius* [49,64]. At the end of hydration, for *L. angustifolius* the total solid loss was recorded as 22 mg/g at 85 °C and 9.6 mg/g at 45 °C. However, for *L. albus* it was recorded as 20 mg/g and 3.5 mg/g at 85 °C and 45 °C, respectively.

Much like hardness, total solid loss also did not follow a linear trend, and the high rates of loss were recorded at elevated temperatures, possibly because at elevated temperatures the structural integrity of cotyledons was affected, as has been elucidated in the preceding section, consequently increasing the mass transfer of intracellular components from legumes to the hydration media [26,29]. For cowpeas, Coffigniez et al. [58] observed a 10-fold increase in dry matter loss as the hydration temperature was increased from 20 to 95 °C. The authors linked this observation to the depolymerisation of cell wall polysaccharides triggered by high temperatures. It is also important to point out that at the beginning of the hydration process, solid loss showed very low temperature dependence (Figure 4c,d). This is evidenced by the fact that up to 50 min, the solid loss values for all the samples were comparable. However, as hydration progressed, solid loss behaviour became increasingly temperature dependent. There was a strong link between the destruction of the cell that is enhanced by high temperatures (85 °C) and molecular mass transfer through the intercellular spaces. Based on our findings comparing solid loss values at 85 °C and 65 °C, it is safe to conclude that hydration at 65 °C for less than 200 min is best for hydration.

### 3.7. Total Phenolics and Anthocyanin Loss during Hydration

Figure 5 show the extent of total phenolic and anthocyanin leaching during hydration of both lupin species. The total polyphenols content (TPC) in raw *L. albus* and *L. angustifolius* was 48.79 mg (GAE)/100g and 79.54 mg (GAE)/100g db, respectively. Our results are close to the TPC values reported for lupins, i.e., from 57 mg (GAE)/100g db to 93.2 mg (GAE)/100g db, by several researchers [2,8,9]. For both lupin species, we observed that high hydration temperature significantly accelerated phenolic content leaching out into the hydration water, while low temperatures hardly had any effect on the same. As seen in Figure 5a,b, TPC losses after 100 min of hydration at 85 °C and 65 °C were recorded as 60% and 50% (db) for *L. albus* and 25% and 8% (db) for *L. angustifolius*, respectively. On the other hand, anthocyanin in *L. angustifolius* seemed more sensitive to hydration time compared to the *L. albus* (Figure 5c,d). For *L. albus*, temperature did not exhibit any discernible effects, except at 25 °C. However, for *L. angustifolius*, loss of anthocyanin increased significantly with increases in temperature from 25 °C to 85 °C. This was due to the seed coat composition of both species. *L. angustifolius* seeds have colourful patches on the seed coat, which contains anthocyanin, whereas the seed coat of *L. albus* is plain yellowish in colour. The anthocyanin content of seeds truly reflected the seed coat colour [67]. Anthocyanin losses of *L. angustifolius* at 85 °C and 65 °C after 100 min of hydration were reported as 38% and 41%, respectively (Figure 5d). However, any further increase did not increase the anthocyanin loss drastically. During hydration of legumes, it is important to minimise the loss of phytochemicals such as phenols and anthocyanins. Based on our results, hydration at 65 °C for less than 100 min is a feasible temperature to preserve most of the phenols and anthocyanins with compared with 85 °C.

### 3.8. Oligosaccharide and Soluble Fibre Loss during Hydration

Raffinose family oligosaccharides (RFOs) are prevalent in legumes and serve as carbon reserves [68]. During the seed development process, RFOs accumulate in legumes and participate and regulate several metabolic activities [69]. Traditionally, these soluble fibres are difficult to digest by intestinal enzymes and are associated with causing flatulence [68]. However, recent research work suggests that probiotic bacteria such as Bifidobacterium can utilise oligosaccharides to convert them into short-chain fatty acids, which can have beneficial effects on human health [70,71]. The effect of hydration parameters on the leaching behaviour of three major RFOs, i.e., raffinose, stachyose, and verbascose, is shown in Figure 6. For *L. albus*, it was observed that the loss of raffinose (Figure 6a) and stachyose (Figure 6c) increased by almost 25 times, whereas verbascose loss (Figure 6b) increased by 20 times as a result of temperature increases from 25 °C to 85 °C. However, in *L. angustifolius*, for the same increase in temperature, the loss of raffinose (Figure 6d), verbascose (Figure 6e), and stachyose (Figure 6f) increased by 10-fold, 70-fold, and 80-fold, respectively. These results agree with the hydration kinetic trends outlined in previous sections. Higher *D_eff_* for *L. angustifolius* may have resulted in higher RFOs loss. Likewise, hydration above 65 °C exponentially increased the loss of RFO in both lupins. For 25 °C and 45 °C, most of the RFOs were preserved within the grain. High temperatures accelerated the mass diffusivity, which may have increased the oligosaccharide leaching out. Matella et al. [72] reported that same hydration temperature can have differential effects on different beans. Furthermore, they identified that high temperature reduces the raffinose and stachyose content in black, red, and navy beans. More recently, Devkota et al. [46] observed similar results after thermal treatment of common beans. Even though hydration at 45 °C and 25 °C preserved most of the phytochemicals and probiotic fibre and maximised the EMC, it is not feasible to hydrate at low temperatures (<45 °C) from an industry point of view. However, based on our findings, it is safe to conclude that hydration at 65 °C for 200 min is the best time temperature combination to hydrate the lupin compared to the nutrient loss at 85 °C.

The lupin hydration water generated at 85 °C and 65 °C was rich in polyphenols, anthocyanin, and prebiotic fibre. Several research studies indicate the beneficial effects of hydration water and associate its consumption with the prevention of diseases. On the contrary, several researchers suggest that soaking water be removed in order to remove antinutritional substances that may leach during hydration and cause more harm than benefit. It appears to be advantageous not to throw away the soaking water; however, we do not endorse its consumption, as further studies are needed to establish its effects on human health.

## 4. Conclusions

Physical, microstructural, and hydration properties of lupin vary significantly based on the varietal differences and hydration temperatures. The study suggests that although high hydration temperatures can accelerate the mass diffusivity, they can have deleterious effect on phytochemicals and dietary fibre content. It was found that high temperatures reduced the equilibrium moisture content and increased the total solid loss and loss of oligosaccharides in both lupin species due to the depolymerization of cell wall polysaccharides. *L. albus* was found to be harder than *L. angustifolius*; however, after hydration, the loss of hardness was higher in *L. albus*. The hardness and solid losses were very closely related to the hydration kinetics.

The Weibull distribution model was found to best represent the lupin hydration behaviour with high accuracy (*R*^2^ > 0.99). Thermodynamic analysis revealed that increasing the temperature also increased the orderliness in the system and reduced the activation enthalpy. The Gibb’s free energy and activation enthalpy values indicated that the lupin hydration process was non-spontaneous and endothermic in nature. Based on our findings, the best time–temperature combination was identified as 200 min at 65 °C, since at these conditions the *D_eff_* values were moderately higher and the bioactive retention was better. These findings are relevant for developing a mechanistic understanding of the hydration process and designing the hydration process for *L. albus* and *L. angustifolius* to achieve the maximum equilibrium moisture content and yield with minimal loss of solids (phytochemicals and prebiotic fibres).

## Figures and Tables

**Figure 1 foods-12-00709-f001:**
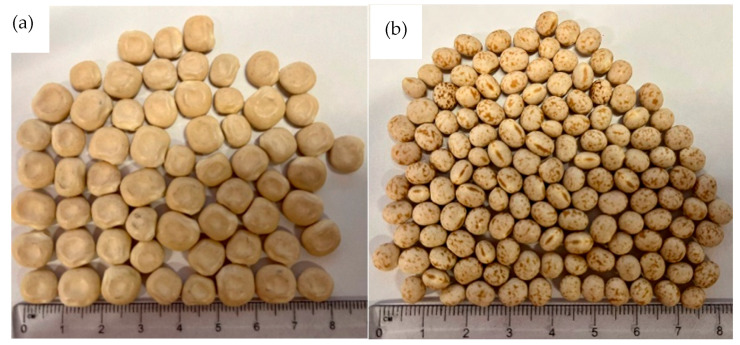
External appearance of Lupins: (**a**) *L. albus*, (**b**) *L. angustifolius*.

**Figure 2 foods-12-00709-f002:**
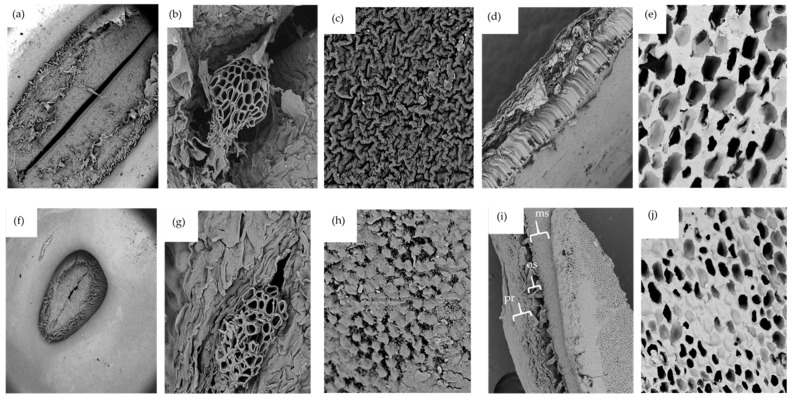
SEM micrographs of *L. albus*: (**a**) hilum (mag 175×), (**b**) micropyle (mag 2300×), (**c**) external surface of the seed coat (mag 3500×), (**d**) transversal cut of seed coat (mag 500×), (**e**) transversal cut of cotyledon (mag 1000×). SEM micrographs of *L. angustifolius*: (**f**) hilum (mag 175×), (**g**) micropyle (mag 2300×), (**h**) external surface of the seed coat (mag 3500×), (**i**) transversal cut of seed coat (mag 500×), (**j**) transversal cut of cotyledon (mag 1000×). Acceleration voltage 10 kV; ms: macrosclereids, os: osteosclereids, pr: paranchyma.

**Figure 3 foods-12-00709-f003:**
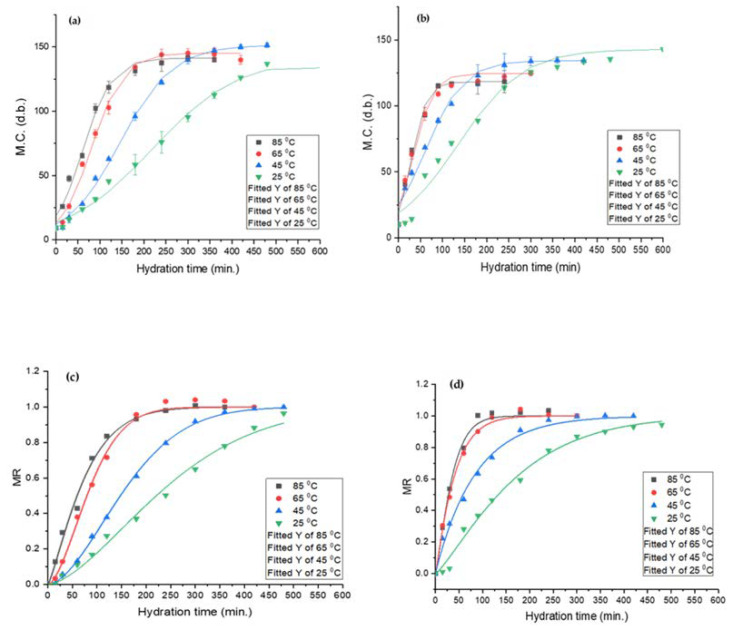
Sigmoidal model (top) and Weibull distribution model (bottom) for *L. albus* (**a**,**c**) and *L. angustifolius* (**b**,**d**).

**Figure 4 foods-12-00709-f004:**
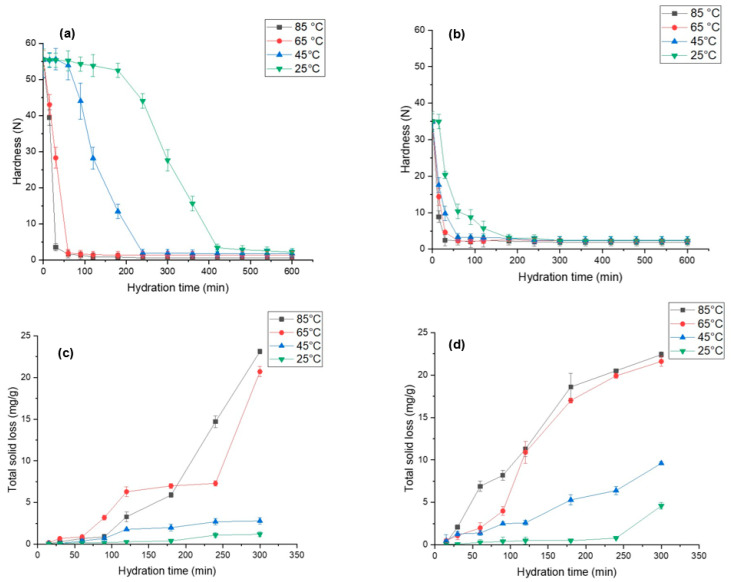
Changes in hardness and total solid loss for (**a**,**c**) *L. albus* and (**b**,**d**) *L. angustifolius* during hydration at different temperature.

**Figure 5 foods-12-00709-f005:**
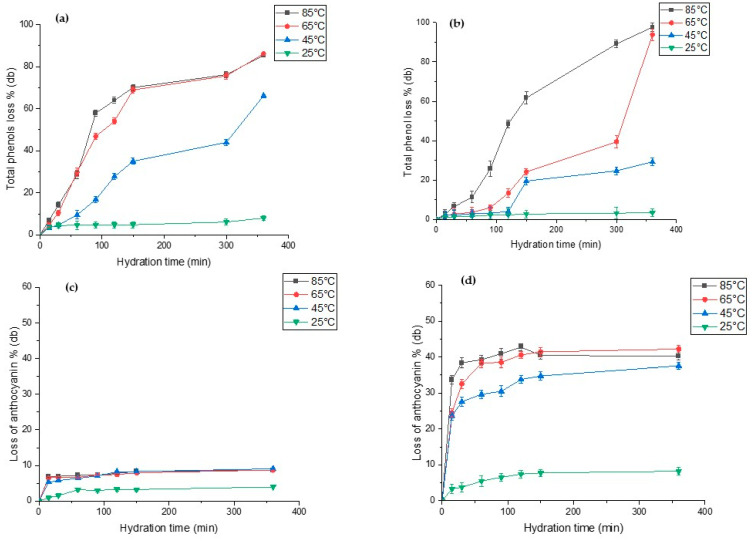
Total phenolic content (%) and total anthocyanin content (%) of hydration of water during hydration at four different temperatures for (**a**,**c**) *L. albus* and (**b**,**d**) *L. angustifolius*.

**Figure 6 foods-12-00709-f006:**
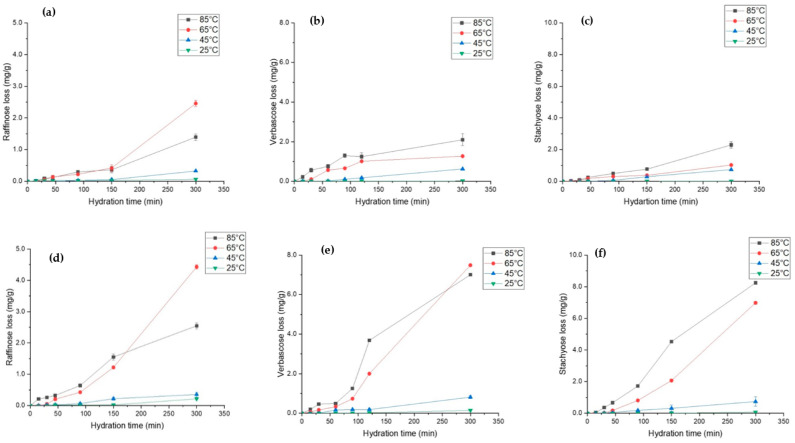
Soluble fibre loss during hydration of *L. albus* at four different temperatures, namely, (**a**) raffinose loss (**b**), verbascose loss, and (**c**) stachyose loss, and for *L. angustifolius*, (**d**) raffinose loss, (**e**) verbascose loss, and (**f**) stachyose loss.

**Table 1 foods-12-00709-t001:** Physical properties of *L. albus* and *L. angustifolius*.

Parameter	*L. albus*	*L. angustifolius*
1000 grain weight (g)	302.03 ± 4.46 ^a^	131.26 ± 3.27 ^b^
Length (L) (mm)	10.79 ± 0.33 ^a^	6.32 ± 0.51 ^b^
Width (W) (mm)	8.78 ± 0.31 ^a^	5.18 ± 0.53 ^b^
Thickness (T) (mm)	5.46 ± 0.30 ^a^	5.03 ± 0.30 ^b^
Geometric mean diameter (mm)	8.02 ± 0.24 ^a^	5.48 ± 0.31 ^b^
Specific surface (mm^2^/mm^3^)	0.78 ± 0.2 ^a^	1.10 ± 0.1 ^b^
Volume (mm^3^)	270.83 ± 1.41 ^a^	86.22 ± 1.23 ^b^
Seed coat thickness (mm)	0.21 ± 0.01 ^a^	0.21 ± 0.01 ^a^

The same letter in the superscript in the same row indicates no significant difference (*p* ≤ 0.05).

**Table 2 foods-12-00709-t002:** Parameter estimates of the Fick model and thermodynamic properties.

Species	Temp (°C)	Effective Diffusivity, *D_eff_* (m^2^/s)	Activation Energy, *E_a_* (kJ/mol)	Enthalpy of Activation, ΔH (kJ/mol)	Entropy of Activation, ΔS (kJ/mol-K)	Gibbs Free Energy, ΔG (kJ/mol)
*Albus*	25	7.41 × 10^−11^	16.00	13.529	−0.224	80.316
	45	1.35 × 10^−10^		13.363	−0.224	84.630
	65	2.10 × 10^−10^		13.196	−0.225	89.282
	85	2.08 × 10^−10^		13.030	−0.225	93.616
*Angustifolius*	25	1.75 × 10^−10^	25.52	23.042	−0.227	90.724
	45	4.38 × 10^−10^		22.875	−0.228	95.415
	65	6.59 × 10^−10^		22.709	−0.229	100.147
	85	1.02 × 10^−9^		22.543	−0.229	104.561

**Table 3 foods-12-00709-t003:** Complied model data.

Model	Species	Hydration Temperature (°C)	Model Parameters		Tests of Fit		
			*k* _1_	*k* _2_	*R* ^2^	Reduced *ϰ*^2^	*RMSE*
Peleg’s model	*L. albus*	25	1.796 ± 0.029	0.004 ± 0.000	0.96	145.34	15.87
		45	2.127 ± 0.030	0.001 ± 0.000	0.96	79.25	11.54
		65	0.849 ± 0.026	0.004 ± 0.000	0.94	44.68	13.40
		85	0.665 ± 0.022	0.005 ± 0.000	0.98	11.33	9.66
	*L. angustifolius*	25	1.367 ± 0.025	0.004 ± 0.000	0.97	32.31	6.98
		45	0.492 ± 0.008	0.006 ± 0.000	0.99	9.84	5.47
		65	0.189 ± 0.006	0.008 ± 0.000	0.94	12.04	8.28
		85	0.387 ± 0.007	0.005 ± 0.000	0.99	9.88	11.54
			** *k* **	** *τ* **			
Sigmoidal model	*L. albus*	25	0.010 ± 0.000	219.997 ± 4.844	0.99	14.87	3.54
		45	0.016 ± 0.000	145.783 ± 2.658	0.99	9.478	2.81
		65	0.028 ± 0.001	82.086 ± 2.523	0.99	19.67	4.01
		85	0.0305 ± 0.002	62.062 ± 2.905	0.98	30.44	4.93
	*L. angustifolius*	25	0.013 ± 0.001	136.627 ± 8.297	0.97	71.63	7.78
		45	0.023 ± 0.002	62.785 ± 3.975	0.98	37.59	5.54
		65	0.043 ± 0.006	33.767 ± 3.193	0.97	49.57	6.20
		85	0.055 ± 0.007	29.038 ± 2.270	0.98	31.04	4.82
			** *α* **	** *β* **			
Weibull distribution model	*L. albus*	25	277.757 ± 6.942	1.598 ± 0.090	0.99	0.00	0.02
		45	182.398 ± 1.643	1.738 ± 0.034	0.99	0.00	0.00
		65	99.131 ± 2.798	1.670 ± 0.114	0.99	0.00	0.02
		85	79.872 ± 3.108	1.259 ± 0.089	0.99	0.00	0.02
	*L. angustifolius*	25	179.335 ± 6.434	1.233 ± 0.074	0.99	0.00	0.03
		45	84.527 ± 3.539	0.993 ± 0.061	0.99	0.00	0.02
		65	41.537 ± 1.625	1.102 ± 0.069	0.99	0.00	0.02
		85	36.971 ± 1.793	1.243 ± 0.107	0.99	0.00	0.02

## Data Availability

The data presented in this study are available on request from the corresponding author.

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
