# Peer review of "Bioactive Nutrient Retention during Thermal-Assisted Hydration of Lupins"

_foods, 2023, doi:10.3390/foods12040709_

Round 1
Reviewer 1 Report
The manuscript reports the study to obtain the best time/temperature combination to hydrate lupins for canning, while minimizing the loss of bioactive nutrients, fibers and total solids during hydration. The work analyzed the hydration and thermodynamic behavior of two species of lupine: L. albus and L. angustifolius, elucidating the effects of the time-temperature combination of hydration at different times during hydration at various temperatures.
The work accurately reports the materials and methods used and the results obtained, skilfully combining graphics and tables. Overall the work is well done but two aspects should be treated better: the practical effects and the bibliographic part:
1) Include in the discussion of the results the biographical reference of the various models used, such as:
Line 285 ref EMC
Line 295 ref fick model.
2) In the conclusions, deepen the practical and applicative implications of these results, given that it is mentioned in the abstract.
Reviewer 2 Report
This study aimed to analyze the hydration and thermodynamic behavior of two lupine species: L. albus and L. angustifolius.
This is an original study, relevant to the scientific community, well designed, and presented.
Given this, small suggestions are presented to the authors for consideration.
Results and discussion
Some excerpts bring a repetition of results already available in the tables. This is most marked between lines 226-235 but is repeated punctually at other points. I request that most of these repetitions be suppressed.
L227. The authors state that genetic factors may be responsible for the variation in seed weight. There is a need to include a reference to this statement. Additionally, this theme should be better explored since a multitude of factors, including those related to the crop factor, could affect grain development.
Some excerpts throughout the discussion lack scientific references. Authors should carefully review all discussions and include them. Among the passages, lines 275 are cited; 460; 464-465; 470-472.
Conclusion
Summarize which treatments are more suitable for use by the population to obtain grains with better technological and chemical qualities.
